# *Fusarium graminearum* Infection Strategy in Wheat Involves a Highly Conserved Genetic Program That Controls the Expression of a Core Effectome

**DOI:** 10.3390/ijms23031914

**Published:** 2022-02-08

**Authors:** Florian Rocher, Tarek Alouane, Géraldine Philippe, Marie-Laure Martin, Philippe Label, Thierry Langin, Ludovic Bonhomme

**Affiliations:** 1UMR 1095 Génétique Diversité Ecophysiologie des Céréales, INRAE, Université Clermont Auvergne, 63000 Clermont-Ferrand, France; florian.rocher63@gmail.com (F.R.); alouane.tarek@gmail.com (T.A.); geraldine.philippe@inrae.fr (G.P.); thierry.langin@inrae.fr (T.L.); 2Institute of Plant Sciences Paris-Saclay (IPS2), CNRS, INRAE, Université Paris-Saclay, Université Evry, 91190 Gif sur Yvette, France; marie_laure.martin@agroparistech.fr; 3Institute of Plant Sciences Paris-Saclay (IPS2), Université de Paris, 91190 Gif sur Yvette, France; 4UMR MIA-Paris, AgroParisTech, INRA, Université Paris-Saclay, 75005 Paris, France; 5UMR 547 Physique et Physiologie Intégratives de l’Arbre en environnement Fluctuant, INRAE, Université Clermont Auvergne, 63178 Aubière, France; philippe.label@inrae.fr

**Keywords:** *Fusarium graminearum*, *Triticum aestivum*, plant–fungus interaction, in planta, transcriptomics, effectoromics, susceptibility factors

## Abstract

*Fusarium graminearum*, the main causal agent of Fusarium Head Blight (FHB), is one of the most damaging pathogens in wheat. Because of the complex organization of wheat resistance to FHB, this pathosystem represents a relevant model to elucidate the molecular mechanisms underlying plant susceptibility and to identify their main drivers, the pathogen’s effectors. Although the *F. graminearum* catalog of effectors has been well characterized at the genome scale, in planta studies are needed to confirm their effective accumulation in host tissues and to identify their role during the infection process. Taking advantage of the genetic variability from both species, a RNAseq-based profiling of gene expression was performed during an infection time course using an aggressive *F. graminearum* strain facing five wheat cultivars of contrasting susceptibility as well as using three strains of contrasting aggressiveness infecting a single susceptible host. Genes coding for secreted proteins and exhibiting significant expression changes along infection progress were selected to identify the effector gene candidates. During its interaction with the five wheat cultivars, 476 effector genes were expressed by the aggressive strain, among which 91% were found in all the infected hosts. Considering three different strains infecting a single susceptible host, 761 effector genes were identified, among which 90% were systematically expressed in the three strains. We revealed a robust *F. graminearum* core effectome of 357 genes expressed in all the hosts and by all the strains that exhibited conserved expression patterns over time. Several wheat compartments were predicted to be targeted by these putative effectors including apoplast, nucleus, chloroplast and mitochondria. Taken together, our results shed light on a highly conserved parasite strategy. They led to the identification of reliable key fungal genes putatively involved in wheat susceptibility to *F. graminearum*, and provided valuable information about their putative targets.

## 1. Introduction

Fusarium Head Blight (FHB), mainly caused by the Ascomycota fungus *Fusarium graminearum*, is one of the most prevalent diseases of small grain cereals, especially in wheat [1,2]. With direct impacts on yield, grain quality and through the accumulation of carcinogenic mycotoxins (e.g., deoxynivalenol, DON) [3,4,5], FHB is considered as a major limiting factor for wheat production in Europe, North America and Asia [6,7,8,9], resulting in substantial economic losses that reached up to USD 1.176 billion over 2015 and 2016 in the USA for instance [10]. Because FHB is expected to be even more frequent and intense along with the rises of temperatures and the occasional increases in air humidity promoted through the climate change [11,12], further research is needed to develop better management strategies and sustainable control solutions [13]. FHB resistance trait is strictly quantitative and involves multiple Quantitative Trait Loci (QTLs) with relatively weak effects [14,15] that makes them insufficient when environmental conditions are favorable to the fungus. Thus, identifying sustainable solutions able to efficiently control FHB epidemics requires the search of alternative sources of resistance. For the last twenty years, the multiple evidences of the role of a plant’s susceptibility factors in promoting pathogen infection have opened new opportunities to identify such pivotal determinants of plant diseases, and a number of studies already reported that mutation or loss of susceptibility genes can be used in resistance breeding [16,17]. With the increasing evidences of the role of wheat’s susceptibility factors in FHB development [18,19,20,21,22,23], elucidating the mechanisms of wheat susceptibility to *F. graminearum* appears as a promising approach to improve FHB resistance [17,24,25,26].

A pathogen’s ability to hijack a host’s biological processes such as defense responses, physiology and primary metabolism to exploit host resources is assumed to be one of the key drivers of a plant’s susceptibility. These interactions involve a complex molecular crosstalk between the two partners, including the delivery of effectors, which include small secreted proteins able to alter host cell structure and to target specific functions into host tissues, the so-called susceptibility factors [27,28,29,30,31,32]. The role of an effector is therefore determined by its in planta localization, i.e., the apoplast or host’s intracellular compartments, and the targeted susceptibility factors [32,33,34,35]. Mining a robust catalog of pathogen effectors, i.e., the effectome, offers major opportunities to improve resistance breeding through the identification of the host’s susceptibility factors, i.e., the targetome. This further could make possible the identification of functional markers to screen plant germplasm, as well as new targets for host-induced gene silencing [32,36,37]. However, their systematic search in silico is still challenging because most of them lack shared protein features or conserved domains within and across species, and very few are structurally characterized [28,32,35]. The only universal fungal effector’s characteristics are their expected secretion and their fine-tuned synthesis along the infection progress [35,38], making in planta exploratory methods such as transcriptomics and proteomics necessary to narrow down the effector candidates and identify the active ones [37,39,40].

Numerous effectors are deployed by pathogens and their role within the molecular crosstalk and in the fate of the interaction is determined by their conservation among pathogen species or between the different strains of a particular species. This conservation is also partly driven by the coevolution with their hosts and their targetome [40,41]. Conserved effectors are thought to play an indispensable role to ensure compatibility by targeting conserved host’s immune or metabolism functions, while specific effectors are thought to be involved in the host’s adaptation and strain aggressiveness [40,41,42,43]. Thus, elucidating the complexity of such molecular crosstalk underlying plant–pathogen interactions and elaborating robust and relevant effectomes require consideration of the diversity from both partners of the interaction. The genomics variability of many fungal pathogen species is well characterized, but its impacts on the infection program remains to be addressed [44,45,46,47,48].

In *Fusarium graminearum*, genomics variability has been well characterized and the first pangenome of the species, built from 20 strains, was published in 2021 [49]. If the in silico characterization of *F. graminearum* secretome [49,50] is now available, our knowledge about the effective in planta effectome remains fragmented and needs to be clarified. Several in planta studies outlined a highly dynamic and complex molecular dialogue between wheat and *F. graminearum*, involving a stage-specific delivery of the effectors [19,25,51,52]. However, the impacts of wheat and *F. graminearum* genetic backgrounds are largely unknown. In a previous proteomics study, *F. graminearum* infection strategy was described in three strains of contrasting aggressiveness facing three wheat cultivars of contrasting susceptibility at one time point, resulting in the identification of highly conserved fungal determinants of the infection [20]. Owing to its higher ability in detecting low-abundant molecules, as are the fungal molecules within host tissues, applying RNA-seq technology over infection progress appears as a promising approach to complete the picture of *F. graminearum* effectome during FHB and to identify its core components.

The presented work addresses qualitative and quantitative characterization of *F. graminearum* paneffectome using an RNAseq-based profiling of the effector coding genes during a time course infection in a two-pronged approach: one using a single aggressive strain facing five wheat cultivars of contrasting susceptibility and a second using three strains of contrasting aggressiveness infecting a single susceptible host. Our results led to the first detailed in planta description of the *F. graminearum* effector repertoire and identified the dynamics of key drivers of the infection process.

## 2. Results

*Fusarium graminearum* transcripts were detected in all infected plant samples with an increase of assigned read pairs along with the infection progress, as expected. In the Host Variability (HostV) experiment, the assignation to *F. graminearum* genome accounted for 1.7%, 8.3% and 16.2% of the assigned pairs at 48 h post-inoculation (hpi), 72 and 96 hpi, respectively (Appendix A). In the Pathogen Variability (PathoV) experiment, it corresponded to 3.5%, 8.6% and 22.6% of the assigned pairs at 48, 72 and 96 hpi, respectively (Appendix A).

### 2.1. In Planta Expression Signature of the F. graminearum Gene Set Coding Secreted Proteins

A total of 9544 and 9898 *F. graminearum* transcripts were reliably identified in the HostV and PathoV experiments, respectively (Appendix A). These two expressed gene sets shared 93% of their accessions. They included 1105 and 1162 genes encoding putative secreted proteins (SP genes) in HostV and PathoV, respectively, accounting for 1183 unique gene accessions (Appendix A). Noticeably, the pangenome mapping approach allowed the identification of 151 new fungal transcripts, absent from the PH-1 reference genome, of which 16 genes were SP genes.

In the HostV data set, 974 SP genes (88.2%) among the 1105 expressed SP genes were expressed in all hosts independently of their susceptibility level, thus representing a host-independent SP gene set (Figure 1A). About 8.7% of the SP gene set (96 genes) was expressed only in some of the different hosts, of which 28% were not detected in the Asian wheat line ‘Chinese Spring’ (CS). A total of 35 genes were identified in only one host of which 43% were specific to the FHB susceptible wheat cultivar ‘Recital’ (REC). A total of six newly characterized gene accessions, not present in the PH-1 reference genome, were detected, including three genes expressed in all the hosts. In the PathoV data set, 1016 SP genes (87.4%) among the expressed SP genes were expressed in all strains independently of their aggressiveness; they form a strain-independent SP gene set (Figure 1B). About 5.5% of the PathoV SP gene set (64 genes) was expressed only by some of the different strains. A total of 82 genes were identified in only one strain and nearly 45% of them were found in the most aggressive strain MDC_Fg1. The PathoV data set resulted in 15 newly identified SP genes of which seven were specifically expressed by MDC_FgU1, four others by MDC_Fg1 and two by MDC_Fg13. SP genes expressed systematically in all the infected hosts and by all the different strains accounted for more than 80% of the whole SP detected genes in both HostV and PathoV experiments (Figure 1C).

### 2.2. Characterization of Differential Expression of SP Gene Sets at the Early Stages of Infection

#### 2.2.1. Fungal Gene Expression in Different Host Cultivars

Among the 3180 HostV differentially expressed genes (DEGs) (Appendix A), a significant enrichment of SP genes was found (*p*-value < 1.13 × 10^−64^), summing 626 accessions significantly impacted by at least one of the tested factors (host, infection progress or their interaction effect) (Figure 2A). They included 201 (32.1%) and 150 (24%) genes whose expression was only impacted by the infection progress (infection regulated genes; Infection-DEGs) and by the host genetic background (host regulated genes; Host-DEGs), respectively, while 204 (32.6%) genes exhibited basal expression differences depending on the infected host cultivar along with differential expression driven by the infection progress (Host + Infection regulated genes; H + I-DEGs). Only 11.3% of the SP DEGs (71 genes) displayed a significant interaction of the two main factors (Host x Infection Progress regulated gene set; HxI-DEGs), depicting differences in expression dynamics in the different host cultivars. Overall, 476 SP genes (76%) were significantly regulated during the infection progress.

#### 2.2.2. Fungal Strain Specificities

Among the 5879 PathoV DEGs (Appendix A), a significant enrichment of SP genes was found (*p*-value < 3.85 × 10^−42^), summing 897 accessions significantly impacted by at least one of the tested factors (strain, infection progress or their interaction effect) (Figure 2B). SP genes whose expression was only impacted by the infection progress (Infection-DEGs) were more than twice as numerous as those only regulated by the strain genetic background (Strain-DEGs) with, respectively, 287 (32%) and 136 (15.2%) genes, while 401 genes (44.7%) exhibited basal expression differences between the different strains along with differential expression driven by the infection progress (Strain + Infection regulated genes; S + I-DEGs). Only 8.1% of the SP DEGs (73 genes) displayed a significant interaction of the two main factors (Strain x Infection Progress regulated genes; SxI-DEGs), depicting differences in expression dynamics depending on the strain. Overall, 761 SP genes (85%) were significantly regulated during the infection progress.

### 2.3. Parsing Secretome Gene Sets towards Effectome Gene Sets That Are Regulated along the Infection Progress

Among the DEGs that proved to be SP genes, those demonstrating fluctuating transcription during the interaction are supposed to be essential components of the infection progress (as effectors are defined [32,35]). With this regard, we established two effectomes in both HostV and PathoV data sets with, respectively, 476 and 761 accessions, by selecting only SP genes with varying expression levels during the infection progress (Appendix A).

In the HostV data set, the aggressive strain MDC_Fg1 displayed 91% (433 genes) of its effectome in all the infected wheat genotypes (Figure 3A). Genes found in only a subset of hosts accounted for 35 genes including two newly identified genes absent from the PH-1 genome while only eight effector genes were host-specific. To gain a better understanding on the impacts of host on fungal gene expression along with the infection progress, we computed a Partial Least Squares Discriminant Analysis (PLS-DA) on the genes expressed in all the hosts (Figure 4A). The first component, explaining 42% of total variance, clearly discriminated the infection stages with the 48 hpi time point on the right side and the 96 hpi time point on the left side of the PLS-DA. For all hosts except CS, the 72 hpi time point marked a transition between the 48 hpi and 96 hpi time-points. The second component, explaining 17% of the variance, discriminated the Asian and spring wheat line CS from the European and winter wheat lines.

In the PathoV data set, the three different *F. graminearum* strains also displayed a conserved effectome gene set, including 90% (682 genes) of the effectome (Figure 3B). As above, effectome genes identified in a subset of strains or in only one strain gathered 39 and 40 genes, respectively; eight of those were newly identified genes absent from the PH-1 genome. The PLS-DA on effectome genes expressed by the three strains showed that the first component, explaining 47% of the variance, also clearly discriminated the stages of infection with the 48 and 96 hpi time points located on the opposite sides of the factorial plane and the 72 hpi time point in between (Figure 4B). The second component of the PLS-DA, explaining 13% of variance, differentiated the most aggressive strain MDC_Fg1 from the two other strains (MDC_Fg13 and MDC_FgU1).

### 2.4. Different Wheat and F. graminearum Genetic Backgrounds Evidenced a Relevant Core Effectome Gene Set Expressed at Specific Infection Stages

As a whole, 357 effector genes were expressed in all the infected hosts by MDC_Fg1 and in Recital by the three strains; they represent an in planta core effectome of *F. graminearum* (Appendix A). Gene and sample Hierarchical Ascendant Clustering (HAC) applied to HostV and Pathov expression data, confirmed the strong impact of infection kinetics on quantitative differences in the core effectome gene set (Figure 5). Genes were expressed per waves and allowed to distinguish the early stages of infection (48 hpi) from the later ones. In comparison with HostV, the PathoV experiment clearly distinguished the intermediate time point (72 hpi) from 48 hpi. More than one-third of the core effectome gene set systematically displayed higher expression levels at 48 hpi as compared to 72 and 96 hpi in both experiments (clusters HostV-1, -2 and clusters PathoV-1, -2) while nearly 60% demonstrated higher expression levels at the latter stages of infection (clusters HostV-4, -5, -6, -7 and clusters PathoV-3, -4, -6). Gathering up to 94% of identical gene accessions, these expression patterns were highly reproducible in both the HostV and PathoV experiments.

### 2.5. Host and Strain Driven Regulations of the Core Effectome Gene Set

#### 2.5.1. Host Cultivar Effects

Basal gene regulations driven by the host genetic background were observed for 43% of the core effectome (154 H + I DEGs; Appendix A) while a difference in gene expression dynamics in different hosts was found for almost 17% of the core effectome (59 HxI DEGs). In addition, the HAC showed that all 72 hpi samples did not cluster together, thus demonstrating that expression patterns can depend on the host (Figure 5A). This exemplifies that according to the infected host, the magnitude of the gene expression increase or decrease could vary. Indeed, the 72 hpi samples of ‘Arche’ (ARC), ‘Courtot’ (COU) and ‘Renan’ (REN) hosts displayed more similarities with most of the 48 hpi samples while the 72 hpi samples of CS and REC hosts were clustered with the 96 hpi samples. These differences were mainly observed in the clusters HostV-1, -2, -4 and -5. For instance, in cluster HostV-4, the gene expression increase was stronger and continuous between 48 and 96 hpi in CS and REC while it mainly took place between 72 and 96 hpi in ARC, COU and REN hosts. Finally, MDC_Fg1 displayed specific gene expression patterns at 96 hpi in CS when compared to the other hosts. In the cluster HostV-2 and -4, MDC_Fg1 gene expression changes were larger when facing CS than the other hosts while in cluster HostV-6 and -7, changes proved to be smaller.

#### 2.5.2. Fungal Strain Effects

Almost 65% of the core effectome gene set displayed strain effects, including 184 S+I DEGs and 46 SxI DEGs (Appendix A). This demonstrates that depending on the strain genetic background, basal gene expression level can be different. Although all 48 hpi samples were clustered together in the HAC, differences at the gene expression level discriminated the three strains (Figure 5B). For instance, clusters PathoV-2 and -3 clearly distinguished MDC_Fg13 and MDC_FgU1 from MDC_Fg1 displaying stronger expression levels while the cluster PathoV-6 clearly distinguished MDC_Fg13 with higher expression levels than the two other strains. The 72 and 96 hpi samples clearly separated MDC_Fg1 from MDC_Fg13 and MDC_FgU1 mainly through the gene clusters PathoV-2, -3 and -4. In cluster PathoV-5, the three strains were clearly discriminated from each other at each time point.

### 2.6. Fusarium graminearum Putative Effectome Displayed Several Targets in Wheat Spikes at Different Infection Stages

A predicted localization within the host was found for 268 proteins (75.07%) of the 357 proteins belonging to the core effectome (Figure 6, Appendix A). Nearly 90% of these proteins displayed a unique predicted localization. Apoplast localization was found for 85% (227 proteins) of the proteins while the others harbored at least one subcellular localization within the host. Both apoplast and cytoplasm localizations were predicted for 29 proteins. The host nucleus was the main subcellular target, gathering 65% (46) of the proteins with a predicted localization. The chloroplast was the second target with 22 proteins while the host mitochondria included only 11 proteins. Multiple predicted intracellular localizations were found for eight proteins. Genes supposed to be secreted in the apoplast or which failed to identify any known subcellular localization signal, displayed mainly increasing expression along with the infection progress. As for the genes coding for proteins supposed to enter host cells, the ratio between genes of increasing and decreasing expression was balanced.

### 2.7. Identification of Additional Effector Features

#### 2.7.1. In Silico Prediction

Based on protein size, net charge and amino acid content [53], EffectorP2.0 led to the identification of 66 putative effectors among the core effectome; 53 genes displayed apoplast localization (Appendix A). Nearly 55% of them showed an increasing expression pattern along with the infection progress while 32% showed a decreasing expression pattern. Three other genes harbored a chloroplast localization signal; all of them were of increasing expression along with the infection progress. Only 1 gene harbored a plant nuclear localization signal and 1 other with a mitochondria transit peptide; both displayed different expression dynamics between the HostV and PathoV experiments. 

#### 2.7.2. Localization in the Fast-Evolving Subgenome

Because previous comparative genomic studies demonstrated that *F. graminearum* putative pathogenicity-related genes were located in variable regions of the genome [49,54,55], the distribution of our core effectome on *F. graminearum* genome was studied (Appendix A). At both inter- and intrachromosomal level, spatial distribution of the core effector genes was uneven (Appendix A). Chromosome 1 accounted for 79 genes, chromosome 2 for 118 genes, chromosome 3 for 94 genes and chromosome 4 for 66 genes. With 13 and 12 genes per Mb, respectively, chromosomes 2 and 3 displayed higher effector gene densities than chromosomes 1 and 4 (6.7 and 7 genes per Mb, respectively). At the intrachromosomal level, genes were preferentially located in the telomeres and in the central regions of the chromosomes. Two-thirds of the core effectome genes were located in the fast-evolving subgenome [55]. While the distribution of intermediate and late expressed genes was relatively balanced on the genome, early expressed genes were mainly located on chromosome 2 (41.5%).

### 2.8. Predicted Functions of the Core Effectome Are Highly Diverse

As a whole, 140 Pfam and 66 GO terms characterizing 230 and 126 genes, respectively, were found in the core effectome (Appendix A). The gene set was significantly enriched in two GO Biological Process terms: ‘carbohydrate metabolic process’ and ‘proteolysis’ with 35 and 15 annotated genes, respectively. At the molecular function level, the core effectome was enriched in ‘metallocarboxypeptidase activity’, ‘hydrolase activity, hydrolyzing O-glycosyl compounds’, ‘polygalacturonase activity’, ‘hydrolase activity’, ‘alpha-L-arabinofuranosidase activity’ and ‘flavin adenine dinucleotide binding’ functions. Finally, the core effectome was enriched in the ‘extracellular region’ as GO Cellular Component term. The search for peptidases in MEROPS database and CAZymes in dbCAN database yielded 38 and 98 genes of the core effectome genes, including 4 and 16 EffectorP-predicted effectors, respectively. Moreover, 36 genes were found in PHI-base, including 22 matches with *F. graminearum*. Fifteen genes were assigned to the ‘unaffected pathogenicity’ category while 16 and 3 other genes were assigned to the ‘reduced virulence’ and ‘reduced virulence_unaffected pathogenicity’ categories, respectively. One gene matched with a *F. graminearum* accession and belonged to the ‘reduced virulence_increased virulence (hypervirulence)’ category. Finally, one gene corresponded to a validated effector that belonged to the category ‘effector (plant avirulence determinant)_increased virulence (hypervirulence)’. Effects on pathogenicity or virulence were shown for 8 EffectorP-predicted effectors.

## 3. Discussion

Taking advantage of different *F. graminearum* strains and wheat host cultivars, this study identified the in planta expression of 840 unique fungal genes harboring effector-associated features (i.e., genes encoding secreted proteins and differentially expressed during the FHB infection), including nine new accessions not included in the reference PH-1 genome. These represent almost 5% of *F. graminearum* pangenome and more than 35% of its putative pansecretome [49].

### 3.1. Fusarium graminearum Infection Involves a Highly Conserved Effectome

Transcriptome profiling of effector coding genes conducted on the three *F. graminearum* strains of contrasting aggressiveness and on the five hosts of contrasting susceptibility to FHB revealed highly conserved effector repertoires. We demonstrated that the three strains shared 90% of their effector-gene transcripts. While at the genomic scale, these three strains shared only 58% of their theoretical secretome [49], our results corroborate a previous in planta proteomics study demonstrating that nearly 100% of the whole identified secreted proteins were accumulated in the same three strains [20]. This emphasizes that the effective infection process of the three strains on wheat is based on a conserved effectome that controls critical plant processes to ensure the success of the infection. Similar results were already found in other pathosystems. For instance, the gene expression analysis of six *Puccinia triticina* strains highlighted a highly conserved infection strategy with 85.7% of the identified secretome genes expressed by all the strains during wheat infection [44]. Similar findings were also reported in the maize–*Exserohilum turcicum* interaction where 97% of the putative effector genes were shared by two strains [48]. Extending this analysis to the role of host genetic background on the expressed effectome, our data also demonstrated a highly conserved infection strategy in different wheat cultivars of contrasting susceptibility to FHB that engages a common effector repertoire shared at 91%. Few hosts’ specific gene expressions were already outlined at the whole-transcriptome scale [56]. Our results support similar conclusions with a special focus on the effectome gene set and are consistent with our previous proteomics study demonstrating the accumulation of the same fungal proteins in different wheat hosts [20]. A core effectome composed of 357 genes expressed by all the strains and in all wheat hosts (Figure 7) exemplified the highly conserved infection program established by *F. graminearum*. Because the interaction is systematically producing FHB disease regardless of the strain aggressiveness or the host susceptibility, these genes likely include key drivers of FHB in bread wheat. Their functions are thus supposed to be crucial determinants of basal processes powering the FHB development in wheat, including 66 putative effector genes and 21 Phi-base matches known to be involved in pathogenicity.

### 3.2. F. graminearum Core Effectors Are Delivered in a Conservative Per-Wave Expression

The core-effectome demonstrated to be deeply remodeled along with the infection progress, depicting the dynamic nature of its components. As previously shown, putative effector proteins were proved to be accumulated at specific stages of the infection process evidencing a specific transition that distinguishes early from late protein accumulations [19]. In line with this previous work, we also observed changing gene expression patterns at the same time (48 to 72 hpi transition), thus corroborating the major reorganization of the molecular arsenal that drives FHB infection. Furthermore, the fine-tuned timing of gene expression was mostly preserved in terms of dynamics for all the strains independently of their aggressiveness and in all the infected hosts independently of their susceptibility level, suggesting that *F. graminearum* set up a widely conserved genetic program with crucial functions required at very precise infection stages (Figure 7). This conserved infection program may be representative of *F. graminearum* generalist lifestyle, i.e., interacting with a wide range of hosts and spreading in different tissues [2,57], which results in a lower selection pressure and coevolution with a specific host species [43,58,59].

Besides these conserved infection patterns, some specific regulations in effector-genes were also found in the different fungal strains, but no clear link between gene expression magnitude and aggressiveness has been observed. Identified effector-genes were mainly located in the fast-evolving part of *F. graminearum* genome, characterized by genes of shorter size, larger variations in exon content and a higher proportion of synonymous and nonsynonymous mutations, together with genes known to be highly transcribed during plant infection in comparison with fungal vegetative growth [49,54,55]. In our study, chromosome 2 for instance, displaying the highest density of polymorphism, also exhibited the highest effector gene density [54]. This polymorphism could explain a part of the observed strain-specific effects on effector-genes expression levels and further protein accumulations. Moreover, intrinsic characteristics of both MDC_Fg1, i.e., a French isolate [60], and ‘Chinese Spring’, i.e., an Asian spring cultivar, might be the main cause of *F. graminearum* specific expression patterns of the late-delivered effectors observed when facing ‘Chinese Spring’ in comparison to the European winter wheat cultivars, suggesting a remarkable ability to adapt to the different molecular contexts expressed in different wheat cultivars.

### 3.3. F. graminearum Infection Strategy Involves Integrative Host Cellular Processes

The search for localization signals within the *F. graminearum* secreted protein sequences revealed that putative effectors can target host apoplast, as well as different subcellular compartments including nucleus, chloroplast and mitochondria at several infection stages (Figure 7). Along with the relatively high diversity of predicted functions (66 GO terms and 140 Pfam), this supports that the infection success is based on a wide array of manipulated host pathways and echoes previous studies that evidenced the diverse nature of processes involved in FHB susceptibility [19,20,23].

Host apoplast appeared as the main target of the *F. graminearum* core effectome, including 53 putative genes with additional effector features, i.e., small cysteine-rich proteins. These genes gathered 65 CAZymes that depict the role of cell-wall degradation during FHB to promote host colonization and nutrient acquisition [61]. Eighteen others belonged to peptidases suggesting that *F. graminearum* is able to override host defense mechanisms especially by interacting with chitin and glucan-triggered immunity and inhibiting host enzymes and proteases as well as to acquire nutrients [62,63,64,65]. Besides these proteases, a guanine-specific ribonuclease was also predicted as a core apoplastic putative effector extending the control of plant stress responses to secreted nucleotidases [61,66]. In addition, three killer toxin KP4-like genes were also identified. Although a previous work has already shown their upregulation during wheat seedling rot disease and FHB, their role in virulence was proved only in seedling rot disease [67].

Intracellular core effectors of *F. graminearum* mainly targeted host nucleus, including two that match with validated virulence factors, a cysteine-rich secretory protein [68] and a PhoD-like phosphatase protein [69,70] along with one gene with additional effector features, i.e., a small cysteine rich protein. Through its eight predicted core nuclear proteases, *F. graminearum* might reprogram host gene expression by interfering with the plant’s transcription factors. This strategy was already found in the pathogenic bacteria *Xanthomonas euvesicatoria* and *Pseudomonas syringae* that target transcription factors involved in phytohormone pathways [71,72]. Nuclear effectors are also known to act on host transcription machinery by a direct binding on DNA, such as the *Melampsora larici-populina* Mlp124478 effector that represses genes involved in defense mechanisms [73]. A same strategy might be involved in the *F. graminearum* infection process though its own nuclear effectors.

Chloroplast and mitochondria were also important targets of *F. graminearum* core effectome, including three and one genes encoding small cysteine rich proteins, respectively, as well as a putative mitochondrial PhoD-like phosphatase virulence factor [69,70]. These organelles represent important biological hubs interconnecting primary metabolism, energy production, signaling pathways and plant responses to stress [74,75]. The inhibition of the defense mechanisms through the manipulation of chloroplast [76] and mitochondrial [77] processes was already evidenced in several plant–fungi interactions and proved here to be part of the *F. graminearum* infection strategy. In the case of the chloroplast, its central role has already been described in previous FHB studies [19,20,23]. Finally, effectors with multiple host targets were also detected, suggesting that one effector can achieve completely different functions during the infection progress. An effector targeting both the chloroplast and the mitochondria was validated in poplar—*Melampsora larici-populina* [78,79] and, as it was outlined in *Blumeria graminis f. sp. hordei* with the BEC1054 RNase-like effector, those versatile effectors seem to disturb one specific process, such as a host’s defense mechanisms, at several levels by interacting with multiple host proteins [80].

## 4. Materials and Methods

### 4.1. Experiments and Biological Material

The current study is based on two experiments, Host Variability (HostV) and Pathogen Variability (PathoV). In HostV, the aggressive strain MDC_Fg1 was inoculated in five bread wheat cultivars, including four French winter genotypes: ‘Arche’ (ARC), ‘Courtot’ (COU), ‘Recital’ (REC), ‘Renan’ (REN) and the Asian spring genotype ‘Chinese Spring’ (CS). In PathoV, three *F. graminearum* strains of contrasting aggressiveness, in decreasing order of aggressiveness MDC_Fg1, MDC_Fg13 and MDC_FgU1 [20] were inoculated in the susceptible wheat cultivar, ‘Recital’ (REC). For both experiments, samples were collected at 48, 72 and 96 h post-inoculation (48, 72 and 96 hpi) (Appendix A).

#### 4.1.1. Preparation of the *Fusarium graminearum* Inoculum

Spores for all the strains were generated according to the same protocol. Mycelium was grown on Potato Dextrose Agar (PDA) medium (39 g of PDA for 1 L of reverse osmosis water) during eight days in the dark at 23 °C. To generate spores, mycelium plugs were suspended in a Mung Bean Broth medium (MBB, 40 g of organic mung bean per L of reverse osmosis water) during seven days in the dark at 23 °C and with an agitation of 150 rpm. Spores were isolated and stored in sterilized water at −20 °C.

#### 4.1.2. Plant Growth Conditions

Wheat seeds were sown in buckets and kept at 20 °C during two weeks for germination. While Chinese spring plantlets were kept at 20 °C, an eight-week vernalization was performed at 4 °C and with a 8:16 (L:D) light cycle for the winter cultivars. Then, all plantlets were transplanted in 4 L pots and transferred into a controlled growth chamber with an automatic watering system. To allow tillering, plants were grown during three weeks at 17 °C/15 °C (day/night), a relative humidity of 70% and a 12:12 light cycle. Then, the experimental conditions were set as follow: 16:8 light cycle with 21 °C/17 °C (day/night) temperatures and a constant relative humidity of 80%.

#### 4.1.3. Experimental Procedures

Complete factorial experiments were designed for HostV and PathoV. For the HostV experiment, the combinations of the five host cultivars and the three time points measured on three biological replicates represent a total of 45 samples. For the PathoV experiment, the combinations of the three strains and the three time points measured on four biological replicates (except for MDC_Fg13 × 72 hpi modality with 3 replicates) represent a total of 35 individuals. Experimental designs were surrounded by additional plants to limit any edge effects. Infection was performed at mid-anthesis by inoculating 10 µL of inoculum at a concentration of 10^5^ spores/mL in the floral cavity of the six central spikelets of three synchronous flowering spikes. For each individual, the inoculated spikelets were collected and immediately placed in liquid nitrogen. Samples were ground in fine powder and stored at −80 °C before RNA extraction.

#### 4.1.4. RNA Extraction and Sequencing

Total RNA was extracted from 100 mg of the frozen powder described above, using a TRIzol protocol (TRI reagent^®^, Sigma-Aldrich, St. Louis, MO, USA), followed by a FastDNase treatment (TURBO DNA-freeTM Kit, Thermo Fisher Scientific, Waltham, MA, USA). Electrophoresis with 1% agarose gel buffered in Tris-Acetate-EDTA was used to control sample quality. An amount of 10 µg of RNA per sample were used for sequencing. cDNA libraries were prepared with the TruSeq Stranded preparation kit reverse oriented (Illumina, San Diego, CA, USA). For HostV samples, 2 × 150 base paired-end sequences were generated using Illumina HiSeq4000 and NovaSeq6000 at the Genoscope, the French National center of sequencing [81]. Sequencing of PathoV samples was performed using Illumina NovaSeq6000 at GeT platform of GenoTOUL [82].

### 4.2. Fusarium graminearum Pangenome Construction and Characterization

To construct the reference *F. graminearum* pangenome, we used all the genomes assembled and publicly accessible. A total of 19 genomes of different origins assembled from this species [49,54,55,83,84] and previously characterized [49] were used along with the reference strain PH-1 genome GCA_900044135.1 assembly [85,86]. The assemblies of three strains (CS3005, DAOM_233423, DAOM_241165) were publicly available. Fourteen strains (INRA-156, INRA-159, INRA-164, INRA-171, INRA-181, YL-1, HN9-1, HN-Z6, MDC_Fg13, MDC_Fg5, MDC_Fg8, MDC_202, MDC_Fg593 and MDC_Fg851) were available as Illumina raw data and were assembled with SPAdes v3.13.0 [87]. The two strains MDC_Fg1 and MDC_FgU1 were available as Pacbio long reads and were assembled with HGAP4 implemented in SMRT Link v5.0 [88]. Then, we performed a mapping of each assembled genome on the PH-1 reference genome using Minimap2 v2.12 [89]. Contigs unmapped and larger than 1 kb in size have been recovered. Additionally, redundant contigs have been removed with Cd-hit v4.6.7 [90]. In this way, we completed the PH-1 reference genome and represented a reference pangenome. On the other hand, to create the GFF/GTF annotation files of the reference assembled pangenome, we used the MAKER2 v2.31.10 annotation pipeline [91] with a combination of evidence-based methods (transcriptome data and homology with known proteomes) and ab initio gene prediction with SNAP v2013-02-16 [92] and Augustus v3.2.1 [93]. The repeating elements of the assembled pangenome were masked using the library of all the repeated elements from the Repbase database (1 February 2017 update) [94] using RepeatMasker v4.0.7 [95]. This *F. graminearum* pangenome is composed of 17,647 protein coding genes, adding 3502 genes to the GCA_900044135.1 assembly of the reference strain PH-1.

The putative pansecretome of *F. graminearum* was generated from the panproteome using SignalP v5.0b software [96] for conventionally secreted proteins and using ApoplastP v1.0.1 [97] for unconventionally apoplastic secreted proteins. This added 437 putative secreted proteins to the reference strain PH-1 and achieved a total of 2339 proteins, including 1656 conventionally secreted proteins. Localizer v1.0.4 [98] was used on protein mature sequences generated by SignalP v5.0b to predict the protein’s subcellular localization within the host. Gene Ontology (GO) and Protein family (Pfam) annotations were generated with Interproscan v5.52-86.0 [99]. To identify putative genes related to pathogenicity, Blastp [100] searches were performed against the Pathogen–Host Interaction database v4-10 (PHI-base) [101,102] and against MEROPS v12.1 for peptidases and their inhibitors [103]. Carbohydrate-Active Enzymes (CAZymes) screening was performed with HMMER v3.1b2 [104] against the dbCAN HMM profile database v9.0 [105]. Secreted proteins with additional effector features, i.e., small cysteine rich proteins, were predicted with EffectorP2.0 software [53].

### 4.3. RNA-Seq Bioinformatic Analysis

The RNA-seq data obtained in both HostV and PathoV experiments were analyzed separately using a dual-genome mapping approach, thus generating two independent data sets (Appendix A). Calculations were performed on the supercomputer facilities of the Mésocentre Clermont Auvergne University [106] and on the TGCC infrastructure of the CEA [107].

#### 4.3.1. Data Cleaning Step

Raw reads were trimmed for adapters and low quality bases (phred score < 20) with TrimGalore v0.6.5 [108]. Noncalled bases and polyA tails were trimmed from reads with homemade Perl scripts. Then, low complexity (compression size < 65%) and short (size < 60 nucleotides) reads were discarded from the fastq files using homemade Perl scripts. Sample files were decontaminated by mapping with STAR v2.7.1.a [109] against a homemade database of potential contaminants composed of 1649 viral genomes, 9267 bacterial genomes and the human genome. The genomes were downloaded from the NCBI Reference Sequence Database [110]. Best whole genome assembly available for each organism or clade without gaps and unlocalized scaffolds were used while genomes with more than ten successive ambiguous bases were discarded. Ribosomal RNA reads were removed with SortMeRNA v4.2.0 [111] against a database of wheat and *F. graminearum* rRNAs built from noncoding RNA genes fasta file of *Triticum aestivum* GCA_900519105.1 assembly [112] and PH-1 GCA_900044135.1 genome assembly [85].

#### 4.3.2. Mapping and Assignation

Genome and annotation files of *Triticum aestivum* v1.1 [113] and *F. graminearum* (pangenome) were combined into a host–pathogen genome. This combined genome contains 269,428 (high confidence and low confidence genes) wheat genes and 17,647 *F. graminearum* genes. The cleaned RNA-seq files were mapped against this combined genome with STAR v2.7.1a. For each species of the pathosystem, gene-level counts were generated with featureCounts software from the subread v2.0.1 package [114]. Only uniquely mapped read pairs were considered at the read count step.

### 4.4. Statistical Analysis of F. graminearum Expression Data

Only *F. graminearum* expression data were considered in this study. HostV and PathoV experiments were statistically analyzed independently (Appendix A). Statistical analysis was conducted on R v3.6.3 [115].

#### 4.4.1. Gene Filtering and Normalization

For both experiments, genes were filtered according to their expression levels. HostV and PathoV genes were filtered per host and per strain, respectively, with a 4 Counts Per Million (CPM) threshold in at least 3 samples independently of the time point. Counts were normalized according to library size with the trimmed mean of M values (TMM) method implemented in edgeR package [116,117].

#### 4.4.2. Differential Expression Analysis

The differential analysis is based on a negative binomial generalized linear model, where the logarithm of the proportion of normalized counts for a gene is modeled by all the factors describing the experiment. For the HostV experiment, the log2 of the average normalized gene expression is an additive function of a time effect (3 modalities), a host effect (5 modalities) and an interaction between the time and the host (15 modalities). For the PathoV experiment, the log2 of the average normalized gene expression is an additive function of a time effect (3 modalities), a strain effect (3 modalities) and an interaction between the two factors (9 modalities).

To make the processing of these two complex multifactorial designs easier, Differential Expression (DE) analysis was made using DiffAnalysis_edgeR function of DiCoExpress [118], which is a script-based tool implemented in R using pre-existing R packages, including edgeR for the differential analysis. This function generates automatically a large number of contrasts. In the HostV experiment, we considered the difference between two hosts at a given time point, the difference between two time points given a host and the interaction effect defined as the difference between two time points given a host minus the difference between the same two time points given a different host. It leads to a total of 77 contrasts. In the PathoV experiment, we considered the difference between two strains at a given time point, the difference between two time points for a given strain and the interaction defined as the difference between two time points given a strain minus the difference between the same two time points given a different strain. It leads to a total of 27 contrasts. For each contrast a likelihood ratio test was applied and raw *p*-values were adjusted with the Benjamini–Hochberg procedure to control the false discovery rate. A gene was declared differentially expressed if its adjusted *p*-value is lower than 0.001. Enrichment in secretome genes between expressed and DE gene sets was performed using a hypergeometric test with the ‘phyper’ function of the R stats package [119] with a *p*-value threshold of 0.01.

#### 4.4.3. Expression Pattern Characterization

For the following analyses, raw count values were normalized using the regularized logarithm transformation (rlog) implemented in the DESeq2 package [120] before being z-score transformed. To assess the capacity of the selected genes to discriminate the samples according to the host or strain × time point combinations and their relevance to describe the infection stages, we performed Partial Least Squares Discriminant Analysis (PLS-DA) using the R package mixOmics [121] with 10 components and default parameters for the other options. Heatmaps representing gene expression patterns were made using the Pheatmap package [122]. The Hierarchical Ascendant Clustering (HAC) of genes and samples was performed with the ward.D2 agglomeration method [123] applied on the Euclidean distance matrices.

#### 4.4.4. Functional Enrichment Analysis

Over-representation of GO terms and Pfam in gene sets was performed using a hypergeometric test with the ’phyper‘ function of the R stats package [119]. The *p*-values were adjusted for multiple testing with Benjamini–Hochberg method and a threshold of 0.01 was applied to select the enriched GO terms and Pfam.

### 4.5. Genomic Localization of Effectome Genes

Genomic localizations of the genes were extracted from the GTF annotation file of PH-1 and their distribution on the chromosomes according to their expression dynamics was generated with the karyoploteR package [124]. Exact positions of the fast and slow subgenome of *F. graminearum* were used to link gene positions and genome structural information [55,125].

## 5. Conclusions

We depicted a detailed map of *F. graminearum* effector genes expressed in planta during FHB infection. They demonstrated that *F. graminearum* displayed a remarkably conserved, fine-tuned and complex infection strategy. This work brings new information about the early stages of wheat–*F. graminearum* interaction and paves the way to further functional analysis of the key fungal molecular drivers of the infection. Taken together with the previous statements made at the proteomic scale on both plant and fungal sides [20,23], our results reinforce the assumption that wheat’s susceptibility to *F. graminearum* is determined by a conserved set of molecular players from both partners. Therefore, core-effectors may lead to the identification of the necessary host susceptibility factors targeted by *F. graminearum* and represent a valuable resource for wheat breeding.

## Figures and Tables

**Figure 1 ijms-23-01914-f001:**
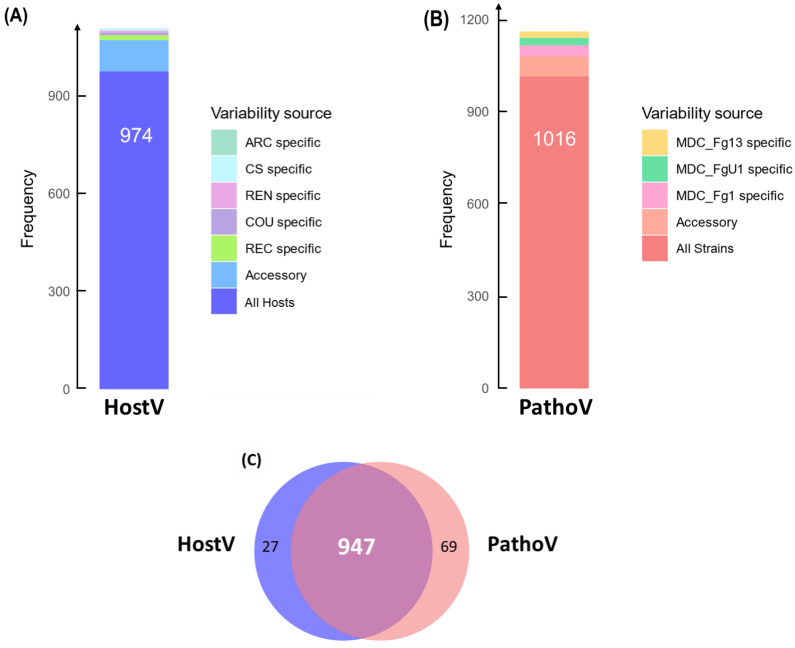
In planta expression signature of *F. graminearum* genes coding for putative secreted proteins. Barplots represent the structure of the gene sets coding for putative secreted proteins expressed in planta for the HostV (**A**) and the PathoV (**B**) experiments. HostV barplot displays the number of genes expressed by the strain MDC_Fg1 in all the infected hosts, in some hosts (Accessory) and in only a specific host: ‘Arche’ (ARC) specific, ‘Courtot’ (COU) specific, ‘Chinese Spring’ (CS) specific, ‘Recital’ (REC) specific or ‘Renan’ (REN) specific. PathoV barplot displays the number of genes expressed by the three strains MDC_Fg1, MDC_Fg13 and MDC_FgU1, by two strains only (Accessory) and by only one strain (MDC_Fg1 specific, MDC_Fg13 specific, MDC_FgU1 specific). The Venn diagram (**C**) represents the intersection of the gene sets expressed in all the hosts (HostV) and expressed in all the strains (PathoV).

**Figure 2 ijms-23-01914-f002:**
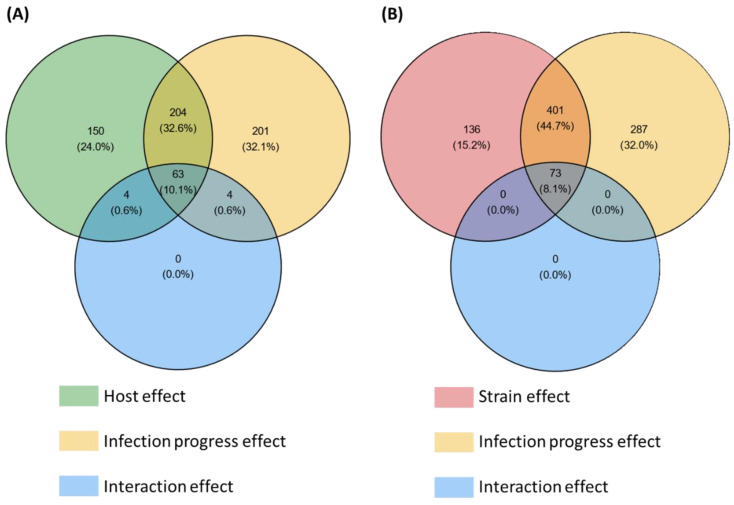
Number of secretome genes from the HostV (**A**) and the PathoV (**B**) experiments significantly impacted by the different effects tested in the differential expression (DE) analysis. For each factor of the DE analysis, the Venn diagrams indicate the number of genes displaying significant expression variations. Significance threshold: *p*-value corrected by Benjamini–Hochberg method < 0.001.

**Figure 3 ijms-23-01914-f003:**
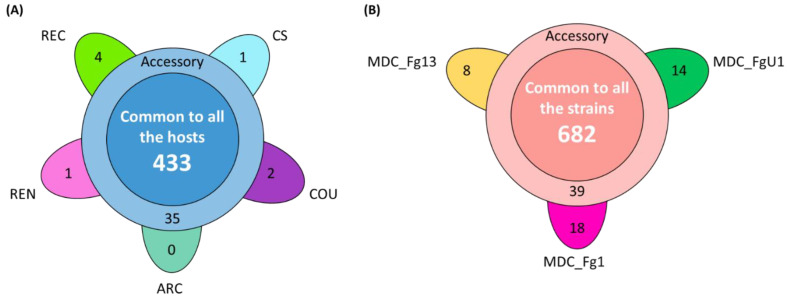
Structure of the HostV (**A**) and PathoV (**B**) effectome gene sets. These sets gather the genes significantly regulated along infection progress and coding for putative secreted proteins. (**A**) The flower plot displays the number of genes expressed by the strain MDC_Fg1 in all the infected hosts (center circle), in some hosts (annulus) and in only a specific host (petals). (**B**) The flower plot displays the number of genes expressed by the three strains MDC_Fg1, MDC_Fg13 and MDC_FgU1 (center circle), by two strains only (annulus) and by only one strain (petals).

**Figure 4 ijms-23-01914-f004:**
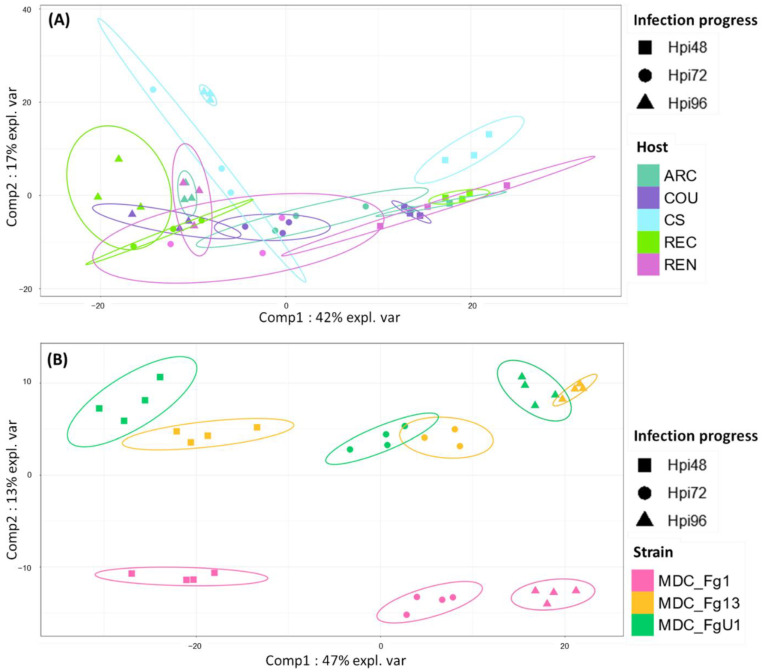
Discrimination of HostV (**A**) and PathoV (**B**) effectome gene sets expressed in all the hosts or by all the strains according to the experimental conditions. PLS-DA method was applied on the 433 genes expressed in all the hosts for HostV (**A**) to predict the host–infection progress combinations and on the 682 genes expressed by all the strains for PathoV (**B**) to predict the strain–infection progress combinations. The plots of the individuals extracted from the PLS-DA are represented on the two first components. For each condition, confidence ellipses are plotted to highlight discrimination strength (level set to 95%).

**Figure 5 ijms-23-01914-f005:**
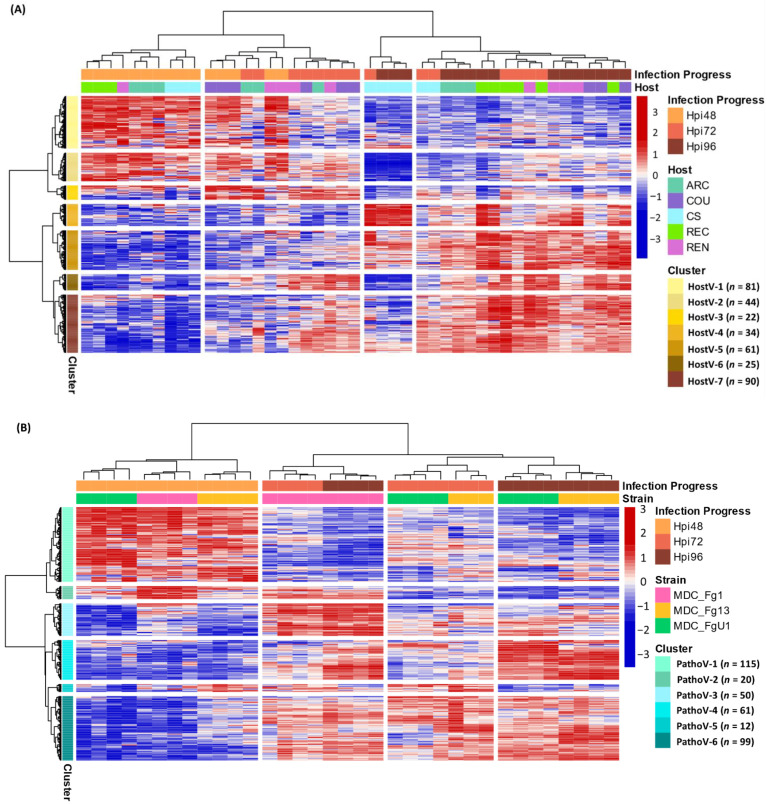
Expression regulation patterns of the core effectome genes along with the infection progress in HostV (**A**) and PathoV (**B**) data sets. The structure of gene and sample data sets were determined by HAC based on Ward’s minimum variance method using the z-score transformed gene expression values. Heatmap color scales represent the z-score transformed expression values of the genes from the core effectome gene set for each sample. The clustering on top of the heatmap represents the experimental conditions which are labeled according to the factors Infection Progress and Host for the HostV experiment, and Infection Progress and Strain for the PathoV experiment. The clustering on the right side of the heatmap represents the genes, which are colored according to their cluster membership.

**Figure 6 ijms-23-01914-f006:**
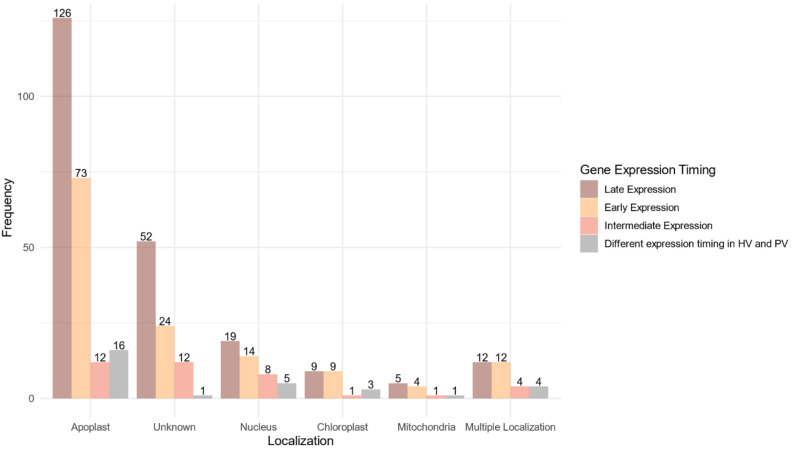
Localization within the host of the fungal proteins encoded by the core effectome gene set according to the expression timing during the infection progress. Barplots represent the frequencies of the predicted localizations within the host of the proteins coded by *F. graminearum* genes expressed at the early stages of the infection (Early Expression), at intermediate stages of infection (Intermediate Expression), at latter stages of infection (Late Expression) in both HostV and PathoV experiments or expressed with different dynamics between the HostV and PathoV experiments.

**Figure 7 ijms-23-01914-f007:**
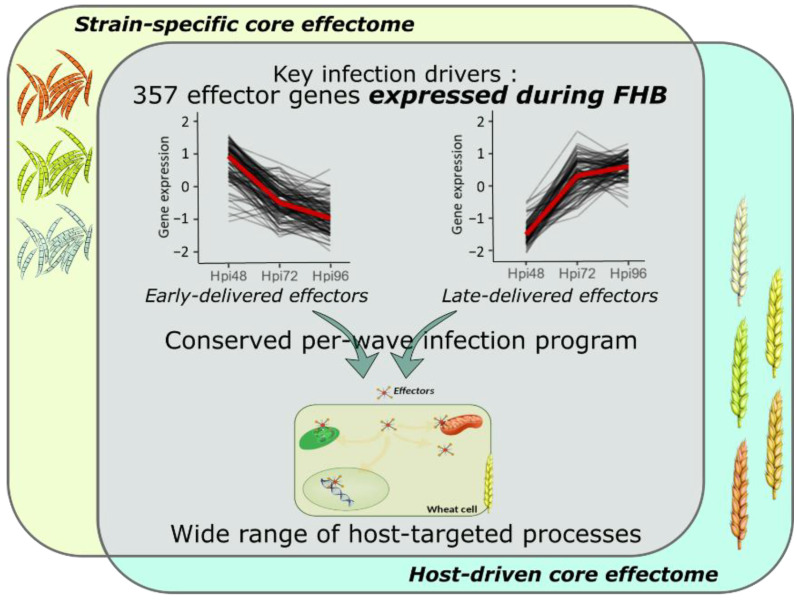
Model summarizing the conserved and complex *F. graminearum* infection strategy on wheat spikes. As a whole, 357 effector genes were identified as the key drivers of FHB infection expressed by all the strains and in all the infected hosts; they represent the *F. graminearum* core effectome. These genes were expressed at very specific infection stages in a per-wave manner, including genes highly expressed at the very beginning of the interaction with the wheat tissues and others highly expressed in the later stages of the infection. The timing of gene expression was mostly conserved independently of the strain or the host. Targeted processes within the host are highly diverse with a wide array of targeted compartments and predicted functions.

## Data Availability

PathoV transcriptomics data on *Fusarium graminearum* that support these findings were deposited in the BioProject repository database, reference number PRJNA795285. HostV transcriptomics data that support the findings were deposited in the BioProject repository database, reference number PRJNA795096. Other data are available in the Appendix A of this article.

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
