# Peer review of "Fusarium graminearum Infection Strategy in Wheat Involves a Highly Conserved Genetic Program That Controls the Expression of a Core Effectome"

_ijms, 2022, doi:10.3390/ijms23031914_

Round 1
Reviewer 1 Report
In this manuscript (ijms-1596039) entitled "Fusarium graminearum infection strategy in wheat involves a highly conserved genetic program that controls the expression of a core effectome" submitted to International Journal of Molecular Sciences, authors analyzed transcriptome of Fusarium graminearum and depicted a detailed expression map of F. graminearum effector genes during infection of bread wheat. They demonstrated that F. graminearum displayed a remarkably conserved, fine-tuned and complex effectome strategy for infection. Overall, this is an interesting work and suitable for publishing on International Journal of Molecular Sciences. The data are convincing and the writing is clear and straightforward.
I just have a few minor points listed below:
- In this study, authors performed a RNAseq based profiling of gene expression during an infection time course using an aggressive F. graminearum strain facing five wheat cultivars of contrasting susceptibility as well as using three strains of contrasting aggressiveness infecting a single susceptible host. I suggest authors to use a picture to display this experiment design in a more readable way. In this picture, key factors such as susceptibility of wheat cultivars, aggressiveness of F. graminearum strains, and infection time course should be indicated.
- Full names of the abbreviations HostV, PathoV, HAC, MDC, ARC, COU, CS, REC, REN, and PLS-DA, should be spelt out at their first appearance in this article.
- Please double-check the reference list. For instance, page number of ‘Human, M.P.; Berger, D.K.; Crampton, B.G. Time-Course RNAseq Reveals Exserohilum Turcicum Effectors and Pathogenicity Determinants. Front. Microbiol. 2020, 11, doi:10.3389/fmicb.2020.00360.’ should be listed (line 844). In addition, abbreviations of ‘MPMI’ and ‘IJMS’ are incorrected listed (line 850, 878, 881, and 911).
Author Response
Response to the reviewer 1:
Comments for Authors
In this manuscript (ijms-1596039) entitled "Fusarium graminearum infection strategy in wheat involves a highly conserved genetic program that controls the expression of a core effectome" submitted to International Journal of Molecular Sciences, authors analyzed transcriptome of Fusarium graminearum and depicted a detailed expression map of F. graminearum effector genes during infection of bread wheat. They demonstrated that F. graminearum displayed a remarkably conserved, fine-tuned and complex effectome strategy for infection. Overall, this is an interesting work and suitable for publishing on International Journal of Molecular Sciences. The data are convincing and the writing is clear and straightforward.
Reply to the reviewer: We thank the reviewer for her/his evaluation and comments that improved our manuscript. We agreed with all the comments and all proposed changes have been amended. Changes can be easily tracked in the manuscript, they have been highlighted using the “track changes” function.
I just have a few minor points listed below:
-
In this study, authors performed a RNAseq based profiling of gene expression during an infection time course using an aggressive F. graminearum strain facing five wheat cultivars of contrasting susceptibility as well as using three strains of contrasting aggressiveness infecting a single susceptible host. I suggest authors to use a picture to display this experiment design in a more readable way. In this picture, key factors such as susceptibility of wheat cultivars, aggressiveness of F. graminearum strains, and infection time course should be indicated.
Reply to the reviewer: As suggested, we had a picture that details our experimental design and the workflow analysis we performed. The scheme was added as Supplementary Figure S4 and referenced in the text of the manuscript (in the “Material and Methods” section).
-
Full names of the abbreviations HostV, PathoV, HAC, MDC, ARC, COU, CS, REC, REN, and PLS-DA, should be spelt out at their first appearance in this article.
Reply to the reviewer: We double-checked all abbreviations and spelt out at their first appearance in the whole manuscript.
-
Please double-check the reference list. For instance, page number of ‘Human, M.P.; Berger, D.K.; Crampton, B.G. Time-Course RNAseq Reveals Exserohilum Turcicum Effectors and Pathogenicity Determinants. Front. Microbiol. 2020, 11, doi:10.3389/fmicb.2020.00360.’ should be listed (line 844). In addition, abbreviations of ‘MPMI’ and ‘IJMS’ are incorrected listed (line 850, 878, 881, and 911).
Reply to the reviewer: We double-checked all the references and amended all typos and abbreviation errors in the list.
Reviewer 2 Report
The authors present the manuscript with the title „Fusarium graminearum infection strategy in wheat involves a highly conserved genetic program that controls the expression of a core effectome”. The manuscript is well organized and the results are well presented. I think the manuscript can be accepted in curent form.
Best regards
Author Response
Response to the reviewer 2:
Comments for Authors:
The authors present the manuscript with the title “Fusarium graminearum infection strategy in wheat involves a highly conserved genetic program that controls the expression of a core effectome”. The manuscript is well organized and the results are well presented. I think the manuscript can be accepted in curent form.
Best regards
Reply to the reviewer: We thank the reviewer for her/his time and her/his evaluation. Best regards.